# Clinical and laboratory factors associated with neonatal sepsis mortality at a major Vietnamese children's hospital

Nguyen Duc Toan[1,2,3], Thomas C. Darton[4], Nguyen Hoang Thien Huong[1,5], Le Thanh Hoang Nhat[2], To Nguyen Thi Nguyen[2], Ha Thanh Tuyen[2], Le Quoc Thinh[1], Nguyen Kien Mau[1], Pham Thi Thanh Tam[2], Cam Ngoc Phuong[6], Le Nguyen Thanh Nhan[1,2], Ngo Ngoc Quang Minh[1], Ngo Minh Xuan[3], Tang Chi Thuong[3,7], Nguyen Thanh Hung[1,3,5], Christine Boinett[8], Stephen Reece[9], Abhilasha Karkey[10], Jeremy N. Day[2,11], Stephen Baker[12,13]*

1 Clinical Departments, Children's Hospital 1, Ho Chi Minh City, Vietnam, 2 Hospital for Tropical Diseases, Wellcome Trust Africa and Asia Programmes, Oxford University Clinical Research Unit, Ho Chi Minh City, Vietnam, 3 Department of Paediatrics, Pham Ngoc Thach University of Medicine, Ho Chi Minh City, Vietnam, 4 Department of Infection, Immunity and Cardiovascular Disease, University of Sheffield Medical School, Sheffield, United Kingdom, 5 Department of Paediatrics, Vietnam National University School of Medicine, Ho Chi Minh City, Vietnam, 6 Hanh Phuc International Hospital, Binh Duong Province, Vietnam, 7 Department of Health, Ho Chi Minh City, Vietnam, 8 Wellcome Trust Sanger Institute, Hinxton, Cambridge, United Kingdom, 9 Kymab, Babraham Research Campus, Cambridge, United Kingdom, 10 Wellcome Trust Africa and Asia Programmes, Oxford University Clinical Research Unit, Kathmandu, Nepal, 11 Centre for Tropical Medicine and Global Health, Nuffield Department of Clinical Medicine, University of Oxford, Oxford, United Kingdom, 12 University of Cambridge School of Clinical Medicine, Cambridge Biomedical Campus, Cambridge, United Kingdom, 13 Department of Medicine, University of Cambridge School of Clinical Medicine, Cambridge Biomedical Campus, Cambridge, United Kingdom

* sgb47@cam.ac.uk

## Abstract

Sepsis is a major cause of neonatal mortality and children born in low- and middle-income countries (LMICs) are at greater risk of severe neonatal infections than those in higher-income countries. Despite this disparity, there are limited contemporaneous data linking the clinical features of neonatal sepsis with outcome in LMICs. Here, we aimed to identify factors associated with mortality from neonatal sepsis in Vietnam. We conducted a prospective, observational study to describe the clinical features, laboratory characteristics, and mortality rate of neonatal sepsis at a major children's hospital in Ho Chi Minh City. All in-patient neonates clinically diagnosed with probable or culture-confirmed sepsis meeting inclusion criteria from January 2017 to June 2018 were enrolled. We performed univariable analysis and logistic regression to identify factors independently associated with mortality. 524 neonates were recruited. Most cases were defined as late-onset neonatal sepsis and were hospital-acquired (91.4% and 73.3%, respectively). The median (IQR) duration of hospital stay was 23 (13–41) days, 344/524 (65.6%) had a positive blood culture (of which 393 non-contaminant organisms were isolated), and 69/524 (13.2%) patients died. Coagulase-negative staphylococci (232/405; 57.3%), *Klebsiella* spp. (28/405; 6.9%), and *Escherichia coli* (27/405; 6.7%) were the most isolated organisms. Sclerema (OR = 11.4), leukopenia <4,000/mm$^3$ (OR = 7.8), thrombocytopenia <100,000/mm$^3$ (OR = 3.7), base excess < –20 mEq/L (OR = 3.6), serum lactate >4 mmol/L (OR = 3.4), extremely low birth weight (OR = 3.2), and

**Data Availability Statement:** All data for this project are available in the paper. Raw data is available upon request.

**Funding:** This project was supported by a Wellcome senior research fellowship to Stephen Baker (215515/Z/19/Z). The funders had no role in the design and conduct of the study; collection, management, analysis, and interpretation of the data; preparation, review, or approval of the manuscript; and decision to submit the manuscript for publication.

**Competing interests:** SR is affiliated with Kymab Ltd. There are no patents, products in development or marketed products to declare. This does not alter our adherence to PLOS ONE policies on sharing data and materials. The authors have declared that no other competing interests exist.

hyperglycaemia >180 mg/dL (OR = 2.6) were all significantly ($p<0.05$) associated with mortality. The identified risk factors can be adopted as prognostic factors for the diagnosis and treatment of neonatal sepsis and enable early risk stratification and interventions appropriate to reduce neonatal sepsis in LMIC settings.

## Introduction

The neonatal period (between birth and the 28[th] day of life) represents an important and vulnerable time as a new-born child adapts to external environments. Infection is a common cause of mortality and morbidity during this period, and sepsis, with an estimated global incidence of 2,202 (95%CI; 1,099–4,360) for every 100,000 livebirths, is associated with a large proportion (11–19%) of this mortality [1]. Consensus definitions and guidelines for sepsis in paediatric populations are widely used by clinicians for goal-based therapeutics [2,3]. However, these guidelines do not address specific conditions associated with neonatal sepsis or its management [4]. Neonatal sepsis can be broadly defined as a clinical syndrome with the presence of systemic inflammatory responses to a suspected or proven infection (classically based on the isolation of bacteria from a blood culture) occurring in children ≤28 days of age [4–7].

Neonatal sepsis is a major cause of hospitalisation and death in low- and middle-income countries (LMICs) [8–10]. Children born in LMICs are at 3-20x fold higher risk of severe neonatal infections than in higher-income countries [11]. In previous studies of neonatal sepsis originating from LMICs in Southeast Asia, the condition was found to be the most prevalent (38%) diagnosis at admission during a 3-year observation period in East Timor [12]. Furthermore, a study conducted in Cambodia from 2007 to 2011 reported an overall mortality of 36.9% for neonatal bloodstream infections [13]. The disease is common in Vietnam and has a high associated mortality [14–16]. A 12-month prospective study performed in a children's hospital in the south of Vietnam described a 16.1% (62/385) mortality rate in neonates with bloodstream infections [14]. A further prospective cohort study conducted in the largest neonatal unit in central Vietnam reported the isolation of a bacterial pathogen from blood in 115/729 (16%) episodes of sepsis among 616 neonatal patients. The overall case fatality rate for microbiologically confirmed sepsis in this study was 46% [15].

Most guidelines for the management of neonatal sepsis have been developed in high-income countries [17–19]. While these guidelines are invaluable in determining optimal care, they are difficult to transpose to less well-resourced healthcare facilities in LMICs [20–22]. Moreover, an insufficient clinical assessment and a delay in the identification of sepsis cases impact on disease management. Additionally, the exact clinical predictors of poor outcomes from neonatal sepsis in LMICs are not well characterised, [21,22]. The few data available suggest that mortality from sepsis was higher in Filipino neonates with abnormalities on chest X-ray [23], and that the co-existence of meningitis or meningoencephalitis was independently associated with mortality in Cambodia [13].

There are limited current data linking the clinical features of neonatal sepsis with the factors associated with mortality in Vietnam. Understanding these factors is essential if we are to identify the most "at-risk" patients early, to develop strategies to improve outcomes. Early recognition of warning signs would allow more timely evaluation, recognition of at-risk patients, the development of a risk-based management approach, and more effective treatment. Here, we aimed to identify the clinical characteristics and laboratory observations associated with mortality in patients with neonatal sepsis in an observational study set in a major children's hospital in Ho Chi Minh City (HCMC) in Vietnam.

## Material and methods

### Ethics statement

Ethical approval for this study was provided by the Oxford Tropical Research Ethics Committee (OxTREC 35–16) and the Ethics Committee of Children's Hospital 1 (CH1) (73/GCN/BVND1). Written informed consent from a parent or guardian was a prerequisite for enrolment into the study. The study was registered with the International Standard Randomised Controlled Trial Number ISRCTN69124914.

### Study design and setting

We conducted a prospective, observational study to characterise the clinical features, to measure the mortality rate, determine the key bacterial pathogens, and identify risk factors associated with mortality of neonatal sepsis at CH1 in HCMC in Vietnam. CH1 is the largest tertiary paediatric centre in Southern Vietnam with 1,400 inpatient beds and >1,600 staff, it receives ~1.5 million outpatient visits and 95,000 admissions each year. The protocol for this study has been published previously [24].

### Inclusion and exclusion criteria

This study included all in-patient neonates (≤28 days of age) at CH1 clinically diagnosed with a primary probable sepsis and receiving a blood culture from January 2017 to June 2018. This center does not provide obstetrical care; therefore, the study participants were out-born neonates. Patients were excluded if informed consent was not provided, if the length of hospital stay was <24 hours, or if death was predicted within 12 hours; these patients were excluded as samples and data were difficult to collect within this time during palliative care and was deemed unethical. Only the primary episode of sepsis was included for each of enrolled neonates, repeat episodes from the same neonate were excluded (patient flow in S1 Fig).

### Study definitions

We used the criteria suggested by the expert meeting on neonatal and paediatric sepsis of European Medicines Agency (EMA) in 2010 for the diagnosis of probable and culture-confirmed sepsis in neonates [25]. A diagnosis of probable sepsis was made when the neonate had ≥2 clinical and ≥2 laboratory signs of sepsis. The patient was latterly diagnosed with culture-confirmed sepsis when there was ≥1 positive blood culture of a presumptive (non-contaminant) pathogen and ≥1 clinical or laboratory sign of sepsis [25].

The clinical signs of neonatal sepsis included: abnormal body temperature (core temperature >38.5°C or <36°C and/or temperature instability); cardiovascular instability (bradycardia [mean heart rate <10th percentile for age in the absence of external vagal stimulus, beta-blockers or congenital heart disease or otherwise unexplained persistent depression over a 0.5–4h time period] or tachycardia [mean heart rate >2SD above normal for age in the absence of external stimulus, chronic unexplained persistent elevation over a 0.5–4h time period] and/or rhythm instability, reduced urinary output (<1 mL/kg/h), hypotension (mean arterial pressure <5th percentile for age), mottled skin, impaired peripheral perfusion); respiratory instability (apnoea episodes or tachypnoea episodes [mean respiratory rate >2SD above normal for age] or increased oxygen requirements or mechanical ventilation requirement); gastrointestinal (feeding intolerance, poor sucking, and abdominal distension); skin and subcutaneous lesions (petechial rash, and sclerema); and non-specific signs (irritability, lethargy, and hypotonia) [25].

The laboratory signs of neonatal sepsis included: white blood cells <4,000/mm$^3$ or >20,000/mm$^3$; immature to total neutrophil ratio (I/T) >0.2; platelet count <100,000/mm$^3$; C-reactive protein (CRP) >15 mg/L; glucose intolerance (hyperglycaemia [blood glucose >180 mg/dL] or hypoglycaemia [blood glucose <45 mg/dL]); and metabolic acidosis (base excess < −10 mEq/L or serum lactate >2 mmol/L) [25].

Early-onset sepsis was defined as the onset of sepsis ≤72 hours after birth [2,26–29]; late-onset sepsis was defined as sepsis developing >72 after birth [2,30–33]. Hospital-acquired sepsis was defined as sepsis occurring ≥48 hours after hospital admission (excluding admission of the mother) [34]; community-acquired sepsis was defined as sepsis arising outside the hospital or <48 hours after hospital admission [8]. Sepsis complicated by organ dysfunction was defined as severe sepsis [5,6] and septic shock as the presence of cardiovascular dysfunction and tissue hypoperfusion [5,6]. The neonatal sepsis attributable death observation was made based on the independent assessments of two qualified neonatologists. The diagnosis of other infectious conditions and comorbidities were based on standard definitions for neonatal-perinatal medicine [35–37]. We identified prematurity as gestational age <37 weeks and extreme prematurity as gestational age <28 weeks. Low birth weight was defined as birth weight <2,500 grams and extremely low birth weight referred to birth weight <1,000 grams.

## Study procedures

One to three milliliters of blood were drawn aseptically before starting antimicrobial therapy from all children and directly injected into BACTEC Peds Plus bottles (Becton Dickinson) before being incubated into an automated BACTEC system at 37˚C for up to five days. Positive bottles were sub-cultured on MacConkey agar, Blood agar and Chocolate agar and incubated at 37˚C for 24–48 hours. Blood culture bottles showing no growth on sub-culture done after 7 days of incubation were reported as negative. Identification of organisms was performed using local protocols using conventional biochemical testing; latterly organisms were referred to a collaborating research laboratory at Oxford University Clinical Research Unit (OUCRU) and confirmed using a mass-spectrometry bacterial identification system (Bruker MALDI-TOF Biotyper). Antimicrobial susceptibility, where performed were interpreted using the most recent CLSI guidelines [38].

Organisms including coryneforms (Corynebacterium, etc), Micrococci, Propionibacterium, Bacillus, alpha haemolytic Streptococci, environmental Gram-negative bacilli, and non-pathogenic Neisseria were considered potential contaminants. The pathogen-contaminant decision was made based on the clinical relevance of the isolated bacteria and the independent assessments by two qualified medical microbiologists. If there was disagreement, then the case was discussed until a decision is reached, in accordance with the study protocol [24].

Patient data were collected on individual case report forms. These data included demographic data, maternal factors, clinical characteristics, laboratory results, diagnoses, treatments, and outcomes. Disease severity was used measured using the Neonatal Therapeutic Intervention Scoring System (NTISS) (39). A study investigator routinely recorded and collected information from the patients.

## Statistical analysis

We performed univariable analysis by using Wilcoxon rank-sum test for continuous variables, and Chi-squared test or Fisher's exact test for categorical variables. All statistical analysis was performed using Stata version 14 and R version 3.4.2, and a significance threshold of $p \leq 0.05$. In the multivariable analysis for mortality outcome, Odds Ratio (OR), 95% Confidence Interval (95% CI), and $p$-values were estimated. Empirical evidence and simulations show that to

avoid overfitting (ideally) the number of predictors should not be larger than the number of deaths divided by 10. There were sixty-nine deaths in this study and thus, we enrolled seven variables into a model and used logistic regression analysis to find factors associated with mortality. The selection of these 7 variables was based on background knowledge and the clinical relevance.

## Results

### Baseline characteristics

Over the 18-month study period from January 2017 to June 2018, 8,497 neonates were admitted to CH1; 524 patients met the clinical criteria for sepsis and were enrolled (Table 1 and S1 Fig). The male to female patient ratio was 1.6:1 and the median age of gestation was 38 weeks (IQR; 33–40 weeks). A high proportion (38.5%) were born prematurely (<37 weeks of gestational age), which was reflected in a similar proportion (42.6%) having a low birth weight. The median birth weight of all patients was 2,700 grams (IQR; 1,800–3,200 grams). The prevalence of Caesarean section delivery was 36.3% (190/524), but other factors often considered to be associated with neonatal sepsis were relatively uncommon (Table 1). Most cases were defined as late-onset neonatal sepsis and were hospital-acquired (91.4% and 73.3%, respectively; Table 1).

In total, 3,091 blood cultures were performed on the 524 sepsis patients, of which 15.2% (469/3,091) were positive; 16.2% (76/469) were determined to be contaminants by local clinical microbiology guidelines and the defined protocol. Ultimately, 192/524 (36.6%) were diagnosed

**Table 1. Baseline characteristics of 524 patients recruited with neonatal sepsis.**

| Variable | n (%) or median (IQR) | |
|---|---|---|
| **Demographic features** | | |
| Male | 320 | (61.1) |
| Female | 204 | (38.9) |
| Gestational age (weeks) | 38 | (33–40) |
| Prematurity | 202 | (38.5) |
| Birth weight (grams) | 2,700 | (1,800–3,200) |
| Low birth weight | 223 | (42.6) |
| **Maternal features** | | |
| Vaginal delivery | 334 | (63.7) |
| C-section delivery | 190 | (36.3) |
| Perinatal asphyxia | 14 | (2.7) |
| Intrapartum fever >38˚C | 4 | (0.8) |
| Suspected/proven chorioamnionitis | 13 | (2.5) |
| Rupture of membranes >18 hours | 8 | (1.5) |
| Maternal hot bed | 57 | (10.9) |
| **Diagnosis of sepsis** | | |
| Probable sepsis | 192 | (36.6) |
| Culture-confirmed sepsis | 332 | (63.4) |
| **Onset of sepsis** | | |
| Early-onset sepsis | 45 | (8.6) |
| Late-onset sepsis | 479 | (91.4) |
| **Acquisition of sepsis** | | |
| Community-acquired sepsis | 140 | (26.7) |
| Hospital-acquired sepsis | 384 | (73.3) |

with probable sepsis and 332/524 (63.4%) were diagnosed with culture-confirmed sepsis. In total, 393 non-contaminant organisms were isolated from the 332 children with a positive blood culture.

## Clinical features

We found that the clinical characteristics of sepsis in this neonatal cohort were highly variable (Table 2). The severity of sepsis was also variable, ranging from moderate to very severe, with 13.9% (73/524) having septic shock. The median (IQR) severity score using NTISS was 17 (11–27), and 399 cases (76.1%) had a NTISS severity score above the low severity threshold of 10. The median (IQR) duration of hospital stay was 23 (13–41) days and death occurred in 69/524 (13.2%) study participants (Table 2).

**Table 2. Key clinical observations of 524 patients recruited with neonatal sepsis.**

| Variable | n (%) or median (IQR) | |
|---|---|---|
| **Clinical features** | | |
| Fever >38.5˚C | 172 | (32.8) |
| Requirement for mechanical ventilation | 191 | (36.5) |
| Impaired peripheral perfusion | 86 | (16.4) |
| Feeding intolerance | 249 | (47.5) |
| Lethargy | 72 | (13.7) |
| **Laboratory results** | | |
| Leukopenia <4,000/mm$^3$ | 29 | (5.5) |
| Leucocytosis >20,000/mm$^3$ | 206 | (39.3) |
| Thrombocytopenia <100,000/mm$^3$ | 135 | (25.8) |
| C-reactive protein (mg/L) | 23.9 | (3.5–58.3) |
| C-reactive protein >15 mg/L | 319 | (60.9) |
| Metabolic acidosis | 182 | (34.7) |
| Base excess (mEq/L) | –15 | (–11.9 to –19.1) |
| Serum lactate (mmol/L) | 4.9 | (2.5–8.7) |
| Serum lactate >4 mmol/L | 33 | (6.3) |
| **Treatment** | | |
| Duration of mechanical ventilation (days) | 5.0 | (2.0–10.0) |
| Duration of total parenteral nutrition (days) | 11.0 | (6.0–21.0) |
| Duration of lipid infusion (days) | 18.0 | (11.0–27.0) |
| Duration of central lines (days) | 21.0 | (7.0–21.0) |
| Surgical/invasive intervention | 171 | (32.6) |
| Shock management | 90 | (17.2) |
| Blood products transfusion | 227 | (43.3) |
| Antacids | 109 | (20.8) |
| Corticosteroids | 13 | (2.5) |
| **Severity** | | |
| Score of severity (NTISS) | 17 | (11–27) |
| Score of severity (NTISS) >10 | 399 | (76.1) |
| Severe sepsis | 121 | (23.1) |
| Septic shock | 73 | (13.9) |
| **Outcomes** | | |
| Duration of stay (days) | 23 | (13–41) |
| Mortality | 69 | (13.2) |

## Laboratory features

Blood samples from all patients were subjected to a complete blood count, C-reactive protein (CRP), and blood culture (Table 2). We found 29/524 (5.5%) cases had leukopenia <4,000/mm$^3$ and 206/524 (39.3%) cases had leucocytosis >20,000/mm$^3$. The median (IQR) CRP concentration was 23.9 (3.5–58.3) mg/L and 319 (60.9%) patients had a CRP >15 mg/L. In addition, 182/524 (34.7%) patients had evidence of metabolic acidosis, which was often severe with a median (IQR) base excess of –15 (–11.9 to –19.1) mEq/L. Lactic acidosis was also observed with 33/524 (6.3%) patients had a value of serum lactate >4 mmol/L, and the median (IQR) serum lactate value being 4.9 (2.5–8.7) mmol/L.

Of the isolated non-contaminant bacteria isolated, 127/393 (32.3%) were Gram-negative and 266/393 (67.8%) were Gram-positive. The most commonly isolated Gram-negative bacteria were *Klebsiella* spp. (28/393; 7.1%), *Escherichia coli* (27/393; 6.8%), *Acinetobacter* spp. (16/393; 4.0%), and *Enterobacter* spp. (14/393; 3.5%) (S2 Table). The most commonly isolated Gram-positive bacteria were coagulase-negative staphylococci (232/393; 58%), *Staphylococcus aureus* (18/393; 4.5%), and *Streptococcus* spp. (10/393, 2.5%) (S2 Table). Notably, we isolated 2 *Streptococcus agalactiae* (group B streptococci) and no *Listeria monocytogenes*.

## Treatment

Empirical antimicrobial treatment with ampicillin, cefotaxime, and gentamycin was initiated when sepsis was suspected on all patients. Requirement for mechanical ventilation was recorded in 191/524 cases (36.5%). The median (IQR) duration of mechanical ventilation and total parenteral nutrition were 5.0 (2.0–10.0) and 11.0 (6.0–21.0) days, respectively. The median (IQR) durations of lipid infusion and central lines were 18.0 (11.0–27.0) and 21.0 (7.0–21.0) days, respectively. A surgical intervention or invasive procedure was performed in 171/524 (32.6%) cases, shock management was implemented in 90/524 (17.2%) cases, and blood product transfusions were given in 227/524 (43.3%) cases (Table 2).

## Factors associated with mortality

The demographic features, maternal factors, clinical characteristics, laboratory results, diagnoses, and treatments hypothesised to impact on mortality are shown in S1 Table. A univariable analysis revealed several factors that were significantly different between the deceased group and the survival group. These factors included demographic features (extreme prematurity and extremely low birth weight), clinical features (hypothermia, temperature instability, reduced urinary output, hypotension, mottled skin, impaired peripheral perfusion, apnoea episodes, increased oxygen requirements, mechanical ventilation, feeding intolerance, poor sucking, abdominal distension, petechial rash, sclerema, lethargy, and hypotonia), category of sepsis (culture-confirmed sepsis, and hospital-acquired sepsis), laboratory variables (leukopenia <4,000/mm$^3$, thrombocytopenia <100,000/mm$^3$, hyperglycaemia >180 mg/dL, base excess <−20 mEq/L, serum lactate >4 mmol/L, electrolyte disturbance, and abnormal coagulation), additional infections, and NTISS disease severity.

To identify key factors associated with mortality we performed a multivariable logistic regression analysis with seven factors: extremely low birth weight; sclerema; leukopenia <4,000/mm$^3$; thrombocytopenia <100,000/mm$^3$; hyperglycaemia >180 mg/dL; base excess <−20 mEq/L; and serum lactate >4 mmol/L. The selection of these seven variables was based on clinical judgement and previous studies [38–46]. We found sequentially that sclerema ($p = 0.005$), leukopenia <4,000/mm$^3$ ($p<0.001$), thrombocytopenia <100,000/mm$^3$ ($p<0.001$), base excess < −20 mEq/L ($p = 0.012$), serum lactate >4 mmol/L ($p = 0.015$),

**Table 3. Multivariable analysis for factors significantly associated with mortality.**

| Mortality-associated factors | OR* | 95% CI* | p-values* |
|---|---|---|---|
| Sclerema | 11.4 | 2.0–63.1 | **0.005** |
| Leukopenia <4,000/mm$^3$ | 7.8 | 2.9–20.8 | **<0.001** |
| Thrombocytopenia <100,000/mm$^3$ | 3.7 | 1.9–7.0 | **<0.001** |
| Base excess < −20 mEq/L | 3.6 | 1.3–9.8 | **0.012** |
| Serum lactate >4 mmol/L | 3.4 | 1.2–9.1 | **0.015** |
| Extremely low birth weight | 3.2 | 1.1–9.0 | **0.022** |
| Hyperglycaemia >180 mg/dL | 2.6 | 1.1–6.1 | **0.021** |

*Logistic regression analysis. OR: odds ratio, CI: confidence interval.

extremely low birth weight (*p* = 0.022), and hyperglycaemia >180 mg/dL (*p* = 0.021) were all significantly associated with mortality at the time of diagnosis (Table 3).

## Discussion

Here we present an observational study of 524 neonates recruited with suspected sepsis recruited at a single centre in southern Vietnam. Sepsis was culture-confirmed in 332 (63.4%) cases. Most of the sepsis cases here were hospital-acquired, suggesting the implementation of a multidimensional strategy to identify risk factors early to allow rapid initiation of clinical intervention. Although the attitude, knowledge, and practices for infection prevention and control have been gradually improved in LMICs, the safeguarding of aseptic hospital environments remains a critical challenge. Sterility is particularly challenging in situations where the prolonged use of vascular lines, parenteral nutrition, and respiratory support cannot be avoided, although active efforts can be made to minimise and to reduce these risks [47].

The prevalence of mortality in this study was 13.2% (69/524 cases), which is lower than smaller studies conducted in the region, specifically in Cambodia (36.9%; 24/65 cases) [13] and (46%; 49/106 cases) elsewhere in Vietnam [15]. The largest single factor associated with mortality was sclerema, which is an uncommon severe panniculitis that manifests as a diffuse hardening of skin and subcutaneous adipose tissue in critically ill, premature, and low-birth-weight infants, particularly during septic shock [41]. Various reports have indicated that severe neonatal sepsis is often associated with sclerema [41,48] and sclerema is in indication of poor outcome. There are limited treatments for sclerema in neonatal sepsis, but intravenous immunoglobulin [49], exchange transfusion [50], and parenteral hydrocortisone may be options [51]. However, given these data we suggest that the focus in LMICs should be how to prevent the 'Pre-sclerema' stage, as it the best indicator of poor prognosis, despite any interventions. Clearly, further studies are needed to determine the clinical applicability of these approaches in a LMIC setting.

In our population, 202 neonates (38.5%) were premature, and 223 (42.6%) had low birth-weight. It has been previously found that the incidence of neonatal sepsis is inversely related to gestational age [52]. Consistently, we found extremely low birth weight to be significantly associated with mortality, which has been identified previously and reported in 25% of early-onset sepsis and 18% of late-onset sepsis [32,38,53]. A cohort study from Mexico found that neonatal sepsis had a mortality rate of 9.5% and low birth weight was a key risk factor [39]. Given the high mortality of sepsis in low-birth-weight neonates, efforts to reduce the incidence and increase the effectiveness of treatment remain an important priority. In any low-birth-weight neonates with suspected sepsis, a combination of ampicillin, cefotaxime, and gentamicin is

empirically initiated. If an organism is isolated and the clinician has the antimicrobial susceptibility data, the treatment is modified. Other important treatments of sepsis in extremely low birth weight new-born infants in our practice include optimizing the oxygen support, maintaining adequate nutrition, fluids, electrolytes, and providing a neutral, stable body temperature.

Our study found that a white blood cell count <4,000/mm$^3$ was significantly associated with death. An investigation of the impact of white blood cell on sepsis also found that a white blood cell <4,000 leads to more severe outcomes [42]. A further study suggested that neonates with white blood cells <5,000 have the highest mortality [43]. Similarly, we found that thrombocytopenia <100,000/mm$^3$ was an important laboratory result in predicting mortality. A cohort study conducted in 460 neonates with sepsis in the Netherlands between 2006 and 2015 showed that thrombocytopenia has a four-fold higher risk of mortality [44]. Additionally, blood glucose >180 mg/dL was identified as a mortality-associated factor. In a study of 1,236 sepsis patients, hyperglycaemia (>200 mg/dL) was associated with simultaneous hyperlactatemia, and hyperglycaemia with concurrent hyperlactatemia (>4 mmol/L) was associated with increased mortality [45]. An adjustment of the glucose infusion rate in parenteral nutrition and insulin administration are effective measures for controlling hyperglycaemia in sepsis patients in our centre. We consider that patients with a value of base excess <–20 mEq/L, and serum lactate >4 mmol/L are the most likely to develop septic shock and cardiorespiratory collapse [46], these patients require prompt intensive care with mechanical ventilation and shock management.

The factors associated with mortality in neonatal sepsis will be applied to routine practice in our centre, particularly in neonates with extremely low birth weight. Sclerema is the only clinical risk factor and our staff have been able to evaluate and recognise early this "red flag" of poor prognosis in neonatal sepsis. Other haematological and biochemical factors including leukopenia <4,000/mm$^3$, thrombocytopenia <100,000/mm$^3$, base excess < –20 mEq/L, serum lactate >4 mmol/L, and hyperglycaemia >180 mg/dL could be identified during the standard work-up in the management of neonatal sepsis. The correction of these abnormalities will contribute greatly to the intensive care of severe sepsis as risk-based therapeutic interventions, and this will help improve the outcome of patients.

Our study has limitations. This work was conducted at the largest referral and tertiary children's hospital in the south of Vietnam; therefore, this may be better resourced than other centres and data collection at a single site may limit the applicability to other hospitals in LMICs in Southeast Asia. Additionally, the interaction between sepsis and other infectious conditions, and the effect of antimicrobial treatment before hospitalization could not be comprehensively investigated. Given local ethical criteria and logistics we also excluded those who were predicted to die within 12 hours of admission. We additionally acknowledge 38.5% of the study participants were born prematurely and had low birth weight, which is likely to have a confounding impact on poor outcome.

We identified the main characteristics associated with mortality in neonatal sepsis in our LMIC setting. These data will enable early risk stratification and interventions appropriate, in addition to guiding the training and education of healthcare professionals in developing more of a risk-based approach to patient care including the importance of infection prevention and control measures. Factors associated with mortality included sclerema, leukopenia <4,000/mm$^3$, thrombocytopenia <100,000/mm$^3$, base excess < –20 mEq/L, serum lactate >4 mmol/L, extremely low birth weight, and hyperglycaemia >180 mg/dL and could be used as predictive variable for the management of neonatal sepsis.

## Supporting information

**S1 Fig. Flow diagram of study recruitment.** Flow diagram outlining patient recruitment from January 2017 to June 2018 and mortality by culture positive and probable sepsis. (TIF)

**S1 Table. Univariable analysis for factors associated with mortality in neonatal sepsis.** (DOCX)

**S2 Table. Profile of 393 bacteria isolated from blood culture in neonatal sepsis.** (DOCX)

## Acknowledgments

We would like to acknowledge all those that were enrolled into the study and their parents and families. We would like to acknowledge our supporting colleagues at CH1 and OUCRU.

## Author Contributions

**Conceptualization:** Nguyen Duc Toan, Stephen Reece, Abhilasha Karkey, Jeremy N. Day, Stephen Baker.

**Data curation:** Le Thanh Hoang Nhat.

**Formal analysis:** Le Thanh Hoang Nhat, To Nguyen Thi Nguyen, Ha Thanh Tuyen, Pham Thi Thanh Tam.

**Funding acquisition:** Stephen Baker.

**Investigation:** Nguyen Duc Toan, Thomas C. Darton, Nguyen Hoang Thien Huong, Ha Thanh Tuyen, Le Quoc Thinh, Nguyen Kien Mau, Cam Ngoc Phuong, Le Nguyen Thanh Nhan, Ngo Ngoc Quang Minh, Ngo Minh Xuan, Tang Chi Thuong, Nguyen Thanh Hung, Christine Boinett, Abhilasha Karkey.

**Methodology:** Nguyen Duc Toan, To Nguyen Thi Nguyen, Ha Thanh Tuyen, Pham Thi Thanh Tam, Stephen Reece.

**Project administration:** Nguyen Hoang Thien Huong, Le Quoc Thinh, Nguyen Kien Mau, Cam Ngoc Phuong, Le Nguyen Thanh Nhan, Ngo Ngoc Quang Minh, Ngo Minh Xuan, Tang Chi Thuong, Nguyen Thanh Hung, Christine Boinett, Stephen Baker.

**Resources:** Nguyen Duc Toan, Thomas C. Darton, Stephen Baker.

**Supervision:** Thomas C. Darton, Christine Boinett, Jeremy N. Day, Stephen Baker.

**Writing – original draft:** Nguyen Duc Toan, Stephen Baker.

**Writing – review & editing:** Thomas C. Darton, Stephen Baker.

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
