## [Decision Letter · Decision Letter 0]

8 Mar 2022

PGPH-D-21-00612

Clinical and laboratory factors associated with neonatal sepsis mortality at a major Vietnamese children’s hospital

Dear Dr. Baker,

Thank you for submitting your manuscript to PLOS Global Public Health. After careful consideration, we feel that it has merit but does not fully meet PLOS Global Public Health’s publication criteria as it currently stands. Therefore, we invite you to submit a revised version of the manuscript that addresses the points raised during the review process.

Please address how you made the distinction between presumptive pathogens and contaminants.

We look forward to receiving your revised manuscript.

Kind regards,

Paddy Ssentongo, MD, PhD, MPH

Academic Editor

Journal Requirements:

1. Please update your Competing Interests statement. If you have no competing interests to declare, please state: “The authors have declared that no competing interests exist.”

2. We notice that your supplementary tables are included in the manuscript file. Please remove them and upload them  with the file type 'Supporting Information'. Please ensure that all Supporting Information files are included correctly and that each one has a legend listed in the manuscript after the references list.

Additional Editor Comments (if provided):

Reviewers' comments:

Reviewer's Responses to Questions

**Comments to the Author**

1. Does this manuscript meet PLOS Global Public Health’s publication criteria? Is the manuscript technically sound, and do the data support the conclusions? The manuscript must describe methodologically and ethically rigorous research with conclusions that are appropriately drawn based on the data presented.

Reviewer #1: Partly

Reviewer #2: Yes

2. Has the statistical analysis been performed appropriately and rigorously?

Reviewer #1: Yes

Reviewer #2: Yes

3. Have the authors made all data underlying the findings in their manuscript fully available (please refer to the Data Availability Statement at the start of the manuscript PDF file)?

Reviewer #1: Yes

Reviewer #2: Yes

4. Is the manuscript presented in an intelligible fashion and written in standard English?

Reviewer #1: Yes

Reviewer #2: Yes

5. Review Comments to the Author

Reviewer #1: Abstract:

Line 44: Denominator for positive cultures is 405. Based the number of subjects and the number of positive cultures, this number should be 524 or 344, respectively. Correct or explain where 405 comes from.

Methods:

line 122: Since the goal of the paper is to identify factors predictive of mortality, it is unclear why the patients at highest risk of death were excluded. Please explain.

line 130: How was a "presumptive pathogen" defined? And elsewhere, how were contaminants defined? There are several organisms included in Table S2 that I would have classified as likely contaminants. Please explain how the distinction between presumptive pathogens and contaminants was made with special attention given to organism that were considered to be pathogens that are typically considered to be contaminants.

line 173 MALDI-TOF is a very expensive piece of equipment. Although Vietnam is classified as a LMIC, the hospital in this paper was obviously well resourced. Babies had access to parenteral nutrition, advanced diagnostics, etc. This is reflected in the high proportion of ELBWs. Their findings will not be generalizeable to most LMIC settings.

Results:

line 204: how were contaminants defined?

line 206: change 34 to 344

line 213: change "mortality was recorded for" to "death occurred in"

line 225: add comma after isolated for readability

line 226: add "commonly" after most

line 228: Reference supplemental table 2

line 258: For the laboratory abnormalities associated with mortality, when are they predictive? Ever? At diagnosis? Another timepoint?

lines 260-265: don't report odds ratios and confidence intervals here since they are in the table

Discussion:

line 331: Add to limitations sections discussion about this hospital being a high-resource hospital.

line 342: Unclear what this sentence means. Would change to indicate that that these factors are associated with mortality in this study.

Reviewer #2: What positively stands out?

This Manuscript presents a scientifically sound topic of significant global public health interest. The Author makes a good case for why this study is important. The title is very specific and does not assume or bias the reader. The Author makes effort to explain the methods used in scientifically sound terms, while avoiding jargon. The methods used are explained in simple terms.

What could be improved?

-It would be great to have a trial profile for this study, so as to follow through how patients were screened and enrolled, with a snapshot of reasons for exclusion.

-Participants who were excluded if death was 'predicted' within the subsequent 12 hours creates investigator selection bias which should be declared under limitations as it impacts on the true picture of neonatal mortality at this facility.

-38.5% of the neonates (study participants) were born prematurely and hence the low birth weight; which also had an impact on poor outcomes. This figure is quite significant and shouldn't be brushed aside. This limits the generalisability of the findings to all neonates. This limitation should be acknowledged as the clinical (and underlying immunological factors and responses to disease) factors for full term neonates may way be very different.

-It's great that clarification was made that selection of the 7 variables to consider in testing for associations was made on clinical judgement; however, most of them were skewed towards the laboratory picture.

-With a high blood culture positivity of 65.6%; it raises the question of what would the impact of treatment received on outcomes

-The recommendations for I.V Immunoglobulins, exchange blood transfusions and parenteral hydrocortisone are very intelligent measures but not practical or for LMIC settings. The global public health issue here would be to focus more on how to prevent the 'Pre-sclerema' stage, since it's presence is already an indicator of poor prognosis; despite the interventions

6. PLOS authors have the option to publish the peer review history of their article (what does this mean?). If published, this will include your full peer review and any attached files.

**Do you want your identity to be public for this peer review?** For information about this choice, including consent withdrawal, please see our Privacy Policy.

Reviewer #1: No

Reviewer #2: No

---

## [Decision Letter · Decision Letter 1]

25 May 2022

PGPH-D-21-00612R1

Clinical and laboratory factors associated with neonatal sepsis mortality at a major Vietnamese children’s hospital

Dear Dr. Baker,

Thank you for submitting your manuscript to PLOS Global Public Health. After careful consideration, we feel that it has merit but does not fully meet PLOS Global Public Health’s publication criteria as it currently stands. Therefore, we invite you to submit a revised version of the manuscript that addresses the points raised during the review process.

From Editor:

This is a critical study, but before I accept the manuscript, please address the reviewers’ comments about the definition of presumptive pathogens, provide the local clinical microbiology guidelines in the methods section or in the supplement and consider carrying out sensitivity analysis where potential contaminants highlighted by the reviewer are excluded. Minor: Please add page and line numbers in the resubmission.

We look forward to receiving your revised manuscript.

Kind regards,

Paddy Ssentongo, MD, PhD, MPH

Academic Editor

Journal Requirements:

1. Please include the full name of the IRB or ethics committee who approved your study.

2. Please update your Competing Interests statement. If you have no competing interests to declare, please state: “The authors have declared that no competing interests exist.”

3. Please provide an Author Summary. This should appear in your manuscript between the Abstract (if applicable) and the Introduction, and should be 150–200 words long. The aim should be to make your findings accessible to a wide audience that includes both scientists and non-scientists. Sample summaries can be found on our website under Submission Guidelines: https://journals.plos.org/globalpublichealth/s/submission-guidelines#loc-parts-of-a-submission

Alternative link: http://journals.plos.org/ploscompbiol/s/submission-guidelines#loc-author-summary

4. We notice that your supplementary tables are included in the manuscript file. Please remove them and upload them with the file type 'Supporting Information'. Please ensure that each Supporting Information file has a legend listed in the manuscript after the references list.

Additional Editor Comments (if provided):

This is a critical study, but before I accept the manuscript, please address the reviewers’ comments about the definition of presumptive pathogens, provide the local clinical microbiology guidelines in the methods section or in the supplement and consider carrying out sensitivity analysis where potential contaminants highlighted by the reviewer are excluded. Minor: please add page and line numbers in the resubmission.

Reviewers' comments:

Reviewer's Responses to Questions

**Comments to the Author**

1. If the authors have adequately addressed your comments raised in a previous round of review and you feel that this manuscript is now acceptable for publication, you may indicate that here to bypass the “Comments to the Author” section, enter your conflict of interest statement in the “Confidential to Editor” section, and submit your "Accept" recommendation.

Reviewer #1: (No Response)

2. Does this manuscript meet PLOS Global Public Health’s publication criteria? Is the manuscript technically sound, and do the data support the conclusions? The manuscript must describe methodologically and ethically rigorous research with conclusions that are appropriately drawn based on the data presented.

Reviewer #1: Yes

3. Has the statistical analysis been performed appropriately and rigorously?

Reviewer #1: Yes

4. Have the authors made all data underlying the findings in their manuscript fully available (please refer to the Data Availability Statement at the start of the manuscript PDF file)?

Reviewer #1: Yes

5. Is the manuscript presented in an intelligible fashion and written in standard English?

Reviewer #1: Yes

6. Review Comments to the Author

Reviewer #1: Methods section: (no page or line numbers were included in the resubmission)

Under study definitions, 1st paragraph, last line: presumptive pathogens are, by definition, non-contaminants. What remains to be clarified is how contaminant vs. presumptive pathogen was defined. The following paragraph states that they were "confirmed using local guidelines" and the results section says they "were determined to be contaminants by local clinical microbiology guidelines". Please provide the guidelines in the methods section. There are many organisms included in Supplemental table 2 which are nearly always contaminants. How were the 4 micrococcus cultures deemed to be pathogenic? For the >100 babies with CoNS, how were these defined to be pathogenic? Others that need justification for their inclusion: Cupriavidus, Halomonas, Ochrobactrum, Paenibacillus, Ralstonia, Roseomonas, Rothia, Kytococcus. These organisms have only rarely been considered to be a human pathogens and are environmental, skin or gut commensals. How do the "local clinical microbiology guidelines" handle these organisms?

This definition has been glossed over by the authors but is vital to the relevance of their paper. If a quarter of the patients they include do not have a plausible putative pathogen identified, they have included many patients who don't truly meet the inclusion criteria and really calls into question the validity of their results.

If they cannot justify why typical contaminants were considered pathogenic in these patients, they should consider removing these patient cases altogether or, at minimum, provide a sensitivity analysis with these cases removed.

7. PLOS authors have the option to publish the peer review history of their article (what does this mean?). If published, this will include your full peer review and any attached files.

**Do you want your identity to be public for this peer review?** For information about this choice, including consent withdrawal, please see our Privacy Policy.

Reviewer #1: No

---

## [Editor Report · Decision Letter 2]

14 Jul 2022

Clinical and laboratory factors associated with neonatal sepsis mortality at a major Vietnamese children’s hospital

PGPH-D-21-00612R2

Dear Professor Baker,

We are pleased to inform you that your manuscript 'Clinical and laboratory factors associated with neonatal sepsis mortality at a major Vietnamese children’s hospital' has been provisionally accepted for publication in PLOS Global Public Health.

Best regards,

Paddy Ssentongo, MD, PhD, MPH

Academic Editor

Even though I couldn't get hold of the reviewer who had wished to take another look at your revisions, I have gone through them and determined that they are sufficient and have therefore recommended the paper to be accepted. Congratulations on such arduous work on neonatal sepsis.